# Transmetalation from Magnesium–NHCs—Convenient Synthesis of Chelating π-Acidic NHC Complexes

Julian Messelberger [1], Annette Grünwald [1], Philipp Stegner [2], Laura Senft [3], Frank W. Heinemann [1] and Dominik Munz [1,*] 

[1] Lehrstuhl für Anorganische und Allgemeine Chemie, Friedrich-Alexander-Universität Erlangen-Nürnberg, Egerlandstr. 1, 91058 Erlangen, Germany; julian.messelberger@fau.de (J.M.); annette.gruenwald@fau.de (A.G.); frank.heinemann@fau.de (F.W.H.)

[2] Lehrstuhl für Anorganische und Metallorganische Chemie, Friedrich-Alexander-Universität Erlangen-Nürnberg, Egerlandstr. 1, 91058 Erlangen, Germany; philipp.stegner@fau.de

[3] Lehrstuhl für Bioanorganische Chemie, Friedrich-Alexander-Universität Erlangen-Nürnberg, Egerlandstr. 1, 91058 Erlangen, Germany; laura.senft@fau.de

\* Correspondence: dominik.munz@fau.de; Tel.: +49-9131-85-27464

**Abstract:** The synthesis of chelating *N*-heterocyclic carbene (NHC) complexes with considerable π-acceptor properties can be a challenging task. This is due to the dimerization of free carbene ligands, the moisture sensitivity of reaction intermediates or reagents, and challenges associated with the workup procedure. Herein, we report a general route using transmetalation from magnesium–NHCs. Notably, this route gives access to transition-metal complexes in quantitative conversion without the formation of byproducts. It therefore produces transition-metal complexes outperforming the conventional routes based on free or lithium-coordinated carbene, silver complexes, or in situ metalation in dimethyl sulfoxide (DMSO). We therefore propose transmetalation from magnesium–NHCs as a convenient and general route to obtain NHC complexes.

**Keywords:** NHC; transmetalation; magnesium; palladium; carbene

## 1. Introduction

*N*-heterocyclic carbene (NHC) ligands have become the powerhouse of modern transition metal chemistry [1–13]. This is largely due to the ease of tuning their electronic and steric properties targeting a specific application. After Arduengo's report on the first crystalline carbene [14], it became common belief that NHCs should be considered strong σ-donor ligands with negligible π-acceptor properties. However, it was later realized that the π-acceptor properties of ancillary carbene ligands are equally important [15]. This led to the design of π-electron-withdrawing carbenes such as Bertrand's cyclic (alkyl)(amino) carbenes (CAACs) [16–21], diamido carbenes (DACs) [22–25], or ferrocenium-decorated NHCs [26–29]. Also, NHCs with saturated backbones (saNHCs, imidazolidin-2-ylidenes) as well as benzannulated congeners (benzNHCs, benzimidazolin-2-ylidenes) show considerable backbonding capabilities. This is either due to the pyramidalization of the amino groups, which reduces the overlap with the π-system within the *N*-heterocycle, or to the π-acidic character of the benzannulated π-system. An excellent example of the importance of these π-effects is represented by conjugated organic singlet biradicaloids derived from carbene scaffolds, where the stability and electronic properties are largely dependent on the nature of the carbene [30–34]. Another outstanding example from transition metal chemistry is the capability of CAACs and saNHCs to stabilize low-valent metal complexes [35].

We showed recently that a strong ligand field as that exerted by carbene ligands with strong σ-donor and strong π-acceptor properties stabilizes multiple-bonded late-transition-metal complexes [36–38]. We therefore became interested in the synthesis of complexes with π-acidic 2,6-pyridine diNHC (CNC) pincer-type ligands derived from the saNHC or benzNHC scaffold bridged by a pyridine moiety [39–41]. In contrast to the ubiquitous conventional imidazoline-derived NHCs, examples for the saturated imidazolidine derivatives are still comparably scarce in the literature [42–44]. This applies even more to CNC pincer-type ligands with saNHC or benzNHC congeners [45–50]. In fact, only one example has been reported for the saNHC derivatives, which is Chirik's iron complex [51,52]. This iron complex was synthesized by in situ deprotonation/metalation using iron hexamethyldisilazane (Fe[N(SiMe$_3$)$_2$]$_2$) as a precursor. Note that this precursor is inconvenient to handle. Besides, because of its high air sensitivity, it has to be distilled prior to use [53,54]. Accordingly, we decided to investigate more expedient and general routes to access metal complexes.

NHC complexes of the s-block metals are still rarely explored [55]. This is also true for magnesium, although the first example of an NHC–magnesium complex dates back to 1993 [56]. Subsequent investigations with s-block metals focused [57–68] largely on anionic ligands because of the weak magnesium–NHC bond [69–74]. For magnesium–NHC complexes, examples of saturated [75] and benzannulated [76,77] NHCs remain extraordinarily scarce. We hypothesized that transmetalation from magnesium–NHCs should be an exceptional mild method to synthesize NHC complexes. In particular, we were hoping for a suppression of carbene-dimerization processes as well as a beneficial template effect by the magnesium metal. Accordingly, we were intrigued by the low cost as well as ease of handling of the related magnesium compounds. Inspiration came especially from two reports in the literature, in which the transmetalation of an anionic NHC ligand to iron [73] and of an acyclic diaminocarbene ligand to copper [78] was reported. In light of the recent report of heavy alkaline earth–NHC complexes embedded in tridentate coordination environments [79], we decided to explore the transmetalation from magnesium–NHCs in more detail. Herein, we report a convenient method for the synthesis of late-transition-metal complexes with tridentate ligands with π-acidic NHCs based on transmetalation from magnesium complexes. Notably, other routes commonly applied [80] failed entirely in our hands or gave low yields and/or impure products. Our results hence suggest transmetalation from magnesium–NHCs as a general and convenient method to access π-acidic and chelating NHC complexes with high yields.

## 2. Results and Discussion

Following our computational predictions [36,37], we decided to synthesize 2,6-pyridine diNHC (CNC) complexes with bulky 2,6-diisopropylphenyl substituents. As NHCs with moderate to fairly strong π-accepting properties, we chose the saNHC (**1$^{sa}$**) and the benzNHC (**1$^{benz}$**) scaffolds. Both ligand precursors were conveniently synthesized by the reaction of 2,6-bromopyridine with 1-(2,6-diisopropylphenyl)-2-imidazoline (benzimidazole, respectively) following a modification of the previously reported procedure (Scheme 1) [52].

**Scheme 1.** Synthesis of ligand precursors; **1$^{sa}$**: *N*-heterocyclic carbene (NHC) with saturated backbone (saNHC), **1$^{benz}$**: benzannulated NHC.

Deprotonation of the carbene precursors with KN(SiMe$_3$)$_2$ led to the clean formation of the free carbenes, as evidenced by $^1$H NMR spectroscopic analysis (Supplementary Materials). The carbenes were stable over the course of days in tetrahydrofuran (THF) or benzene solutions. Unfortunately,

we repeatedly obtained a mixture of compounds when treating the solutions of the free carbene **1$^{sa}$** with metal precursors such as dichloro(1,5-cyclooctadiene)palladium(II) [Pd(COD)Cl$_2$] (Scheme 2; cf. vide infra, Table 1). In fact, reaction control by $^1$H NMR suggested a crude yield of only 18% of the desired, chelated palladium complex for the reaction in THF. When running the reaction in benzene, we observed the formation of the palladium complex with 40% yield.

**Scheme 2.** The free carbenes do not produce a clean reaction with a common palladium precursor such as dichloro(1,5-cyclooctadiene)palladium(II) ([Pd(COD)Cl$_2$]).

**Table 1.** Product selectivity using different solvents and transmetalating reagents.

| Base | Solvent | Crude Product Selectivity |
|---|---|---|
| LiN(SiMe$_3$)$_2$ | THF | ≈60% |
| LiN(SiMe$_3$)$_2$ | benzene | ≈60% |
| KN(SiMe$_3$)$_2$ | THF | ≈40% |
| KN(SiMe$_3$)$_2$ | benzene | ≈40% |
| MgBr$_2$/KN(SiMe$_3$)$_2$ | THF | ≈80% |
| Mg[N(SiMe$_3$)$_2$]$_2$ | benzene | ≈100% |

Upon treatment of **1$^{sa}$** with LiN(SiMe$_3$)$_2$, we observed the quantitative formation of HN(SiMe$_3$)$_2$ with concomitant precipitation of an orange compound, which we believed to be the lithium complex. However, treatment of this compound with [Pd(COD)Cl$_2$] did not result in the clean conversion to the chelated palladium complex (selectivity for the desired palladium complex in THF or benzene ≈ 60%). Treating the free carbene **2$^{benz}$** in THF with [Pd(COD)Cl$_2$] afforded equally a mixture of compounds (≈40% crude selectivity according to $^1$H NMR), with the concomitant strong formation of palladium black.

We subsequently evaluated two other commonly applied synthetic routes to generate NHC complexes. These rely either on the transmetalation from coinage metal complexes generated upon addition of a coinage metal(I) oxide to the conjugated acid of the carbene, or on the in situ metalation by Pd(OAc)$_2$ in dimethyl sulfoxide (DMSO). Whereas the latter method worked well with **2$^{benz}$** (see Section 3.7), we did not obtain satisfactory results for **2$^{sa}$**. Notably, the attempted metalation of **1$^{sa}$** using Ag$_2$O followed by the addition of [Pd(COD)Cl$_2$] consistently gave an impure product mixture due to the supposed hydrolysis of the saNHC scaffold (Scheme 3I). Although the isolation of the (moderately light-sensitive) silver complex might have remedied this issue, we concluded that this route is neither convenient nor time- or cost-efficient.

The reaction with Pd(OAc)$_2$ in DMSO evolved to be very unpractical for **3$^{sa}$**. At room temperature, the reaction proceeded extremely slow (three weeks) and still gave a slight amount of palladium black byproduct, which necessitated the addition of an excess of palladium acetate precursor (Scheme 3II). Furthermore, the purification of the product proved challenging because of the charged character of the product and supposedly the unidentified byproducts. Upon elevating the temperature to 40 °C, we, however, observed not only a faster reaction but also a strongly reduced product selectivity and the strong formation of palladium black. The quantitative removal of DMSO evolved to be challenging as well and could not be quantitatively accomplished through either washing with diethyl ether or heating overnight to 190 °C.

We consequently evaluated the generation of magnesium–NHC complexes for subsequent transmetalation. Indeed, upon addition of magnesium bromide to the in situ generated **1$^{sa}$** and **1$^{benz}$**,

the clean and quantitative conversion to the magnesium complexes **4<sup>sa</sup>** and **4<sup>benz</sup>** was evidenced (Scheme 4).

**I. Transmetalation from Silver Carbene Complex**

1.) Ag₂O
2.) [Pd(COD)Cl₂]
✗
DMSO

**1<sup>sa</sup>**

**II. In-situ metalation by Pd(OAc)₂**

1.1 eq. Pd(OAc)₂
DMSO, rt
**3 weeks**

**1<sup>sa</sup>**

**3<sup>sa</sup>** (impure)

**Scheme 3.** Neither the "silver oxide" transmetalation route (**I**) nor metalation in dimethyl sulfoxide (DMSO) by palladium acetate (**II**) are suitable for the synthesis of complexes with an imidazolidine-based ligand.

1.) 2 eq. KHMDS
2.) MgBr₂
Et₂O/THF

**1<sup>sa</sup>**
**1<sup>benz</sup>**

**4<sup>sa</sup>** (quantitative)
**4<sup>benz</sup>** (quantitative)

**Scheme 4.** Convenient synthesis of magnesium complexes through complexation of MgBr₂ (L = THF).

These complexes could be isolated in quantitative yields and coordinated either THF or pyridine upon dissolution in the latter. The magnesium complexes **2<sup>sa</sup>** and **2<sup>benz</sup>** could also be obtained straightforwardly by treatment of the salt precursors with Mg[N(SiMe₃)₂]₂ in benzene. In this case, no further coordinated solvent molecules seemed to be present, as judged from the ¹H NMR spectroscopic analysis in pyridine-D₅.

Single crystals suitable for the determination of the solid-state structure could be obtained for **4<sup>sa</sup>** through vapor diffusion of pentane into a saturated solution of pyridine (Figure 1).

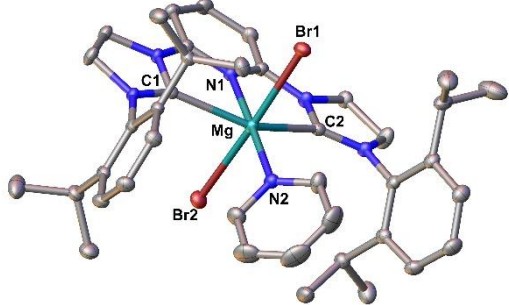

**Figure 1.** Solid-state structure of **4<sup>sa</sup>**. Ellipsoids are shown at the 50% probability level; solvent molecules and hydrogen atoms are omitted for clarity. Selected bond lengths [Å] and angles [°]: N1–Mg 2.162(2), C1–Mg 2.311(2), C2–Mg 2.301(2), Br1–Mg 2.6502(6), Br2–Mg 2.7074(6), N2–Mg 2.171(2); C1–Mg–N 89.27(1), N1–Mg–Br1 73.14(7).

Complex **4^sa** crystallized with a distorted pseudo-octahedral coordination geometry with the two bromido ligands in the apical positions and a coordinated pyridine molecule in the equatorial position. The magnesium–carbene bonds appeared unusually long (C1–Mg 2.311(2) Å, C2–Mg 2.301(2) Å), exceeding all previously reported magnesium carbene complexes in length [61,67,70]. Contrarily, the bond length between the central pyridine moiety and the metal (N1–Mg 2.162(2) Å) appeared to be in the common range for pyridine-coordinated magnesium complexes, as was also found for the coordinate pyridine molecule (N2–Mg 2.171(2) Å).

To our delight, **4^sa** and **4^benz** evolved to be excellent transmetalating reagents. Treatment of a solution of **4^sa** and **4^benz** in THF with either [Pd(COD)Br₂] or FeBr₂ led to the rapid formation of suspensions due to the formation of the desired transition metal complexes. Preliminary investigations with nickel(II) bromide, cobalt(II) bromide, bismuth(III) chloride, and lead(II) bromide indicated that the transmetallation is also feasible with these metal precursors.

The products **3^sa**, **3^benz**, and **5^sa** could be obtained analytically pure after extraction with dichloromethane and subsequent precipitation using diethyl ether (**3^sa**, **3^benz**) or washing with diethyl ether (**5^sa**) in quantitative (**3^sa**), 85% (**3^benz**), and 98% (**5^sa**) yields (Scheme 5). Surprisingly, **3^benz** hydrolyzed slowly in wet solvents, whereas **3^sa** was perfectly stable also in the presence of a large excess of water. The identity of the palladium complex **3^benz** was therefore confirmed by a solid-state structure (Figure 2). The structural parameters of the complex were found to be well in line with those in previous reports on related conventional NHC complexes [41]. All three aromatic rings of the pyridine and benzannulated heterocycle are positioned in one plane with a distance between C3 and C4 (C5, C6, respectively) of 3.352 Å (3.329 Å, respectively). Thus, steric stress of the hydrogen atoms bound to C3, C4, C5, and C6 might be responsible for the sensitivity of **3^benz** to water, which was not found for **3^sa**.

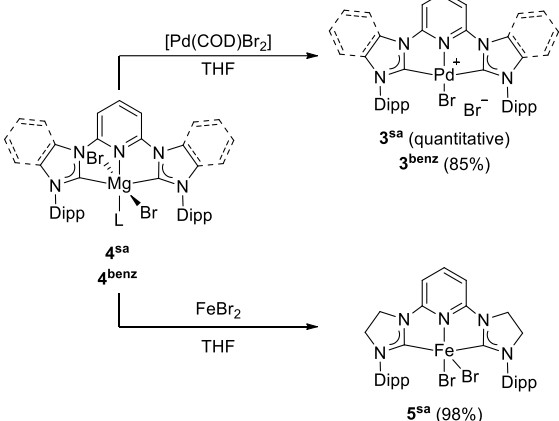

**Scheme 5.** High-yielding and expedient formation of palladium and iron complexes through transmetalation from magnesium complexes.

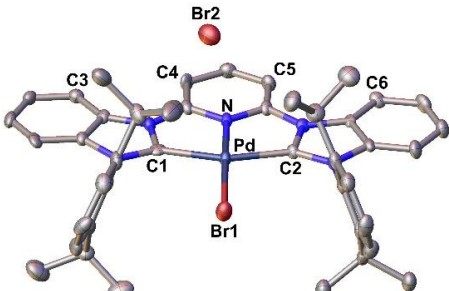

**Figure 2.** Solid-state structure of **3^benz**. Ellipsoids are shown at the 50% probability level. Hydrogen atoms are omitted for clarity. Selected bond lengths [Å] and angles [°]: C1–Pd 2.012(7), C2–Pd 2.027(7), N(pyridine)–Pd 1.977(6), Br1–Pd 2.393(1), C3–C4 3.352, C5–C6 3.329; C1–Pd–N(pyridine) 79.5(3).

## 3. Materials and Methods

### 3.1. General Information

All air-sensitive compounds were synthesized using Schlenk techniques or an MBraun dinitrogen filled glovebox (Mbraun, Garching, Germany). $^1$H and $^{13}$C NMR spectra were recorded on a JEOL ECX 270, a JEOL ECX 400 (JEOL, Freising, Germany), Bruker Avance III HD 600 MHz, or a Bruker AVANCE DRX400 WB instrument (Bruker, Rheinstetten, Germany) operating at 269.71, 399.79, and 600.13 MHz for $^1$H and at 67.82 MHz, 100.62 MHz, and 150.90 MHz for $^{13}$C, respectively, and at a probe temperature of 23 °C. The solvent residual signals were used as an internal reference for the $^1$H NMR and $^{13}$C NMR spectra [81]. NMR multiplicities are abbreviated as follows: s = singlet, d = doublet, t = triplet, q = quartet, spt = septet, m = multiplet. All coupling constants *J* are given in Hertz. Solvents were purified using a two-column solid-state purification system (Glass Contour System, Irvine, CA, USA). Pentane, hexanes, benzene, and toluene were stored over a mirror of potassium; all other solvents were stored over activated molecular sieves. NMR solvents were obtained dry, packaged under argon, and stored over activated molecular sieves or a mirror of potassium ($C_6D_6$). Electrospray-ionization MS (ESI–MS) measurements were performed on a UHR-TOF Bruker Daltonik (Bremen, Germany) maXis plus, equipped with an ESI–quadrupole time-of-flight (qToF) mass spectrometer capable of resolution of at least 60.000 FWHM. Detection was in positive-ion mode, and the source voltage was 3.2 kV. The flow rates were 180 µL/h. The drying gas ($N_2$), to aid solvent removal, was held at 180 °C, and the spray gas was held at 20 °C. The machine was calibrated prior to every experiment via direct infusion of the Agilent ESI–TOF low-concentration tuning mixture, which provided an *m/z* range of singly charged peaks up to 2700 Da in both ion modes. The melting points were determined using a Bibby Scientific SMP10 melting point apparatus. The compounds 1-(2,6-diisopropylphenyl)benzimidazole, 1-(2,6-diisopropylphenyl)imidazoline, dibromo(1,5-cyclooctadiene)palladium(II), dichloro(1,5-cyclooctadiene)palladium(II), $MgBr_2$, and magnesium bis(trimethylsilyl)amide were synthesized according to literature procedures [52,82–87]. All other reagents were obtained from commercial sources and used without further purification.

### 3.2. Synthesis and Characterization of Imidazolinium Salt $1^{sa}$

This compound was synthesized according to a modified literature procedure [52]. An ACE pressure tube was loaded with 2,6-dibromopyridine (581 mg, 2.45 mmol, 1.0 eq.) and 1-(2,6-diisopropylphenyl)imidazoline (1.30 g, 5.64 mmol, 2.3 eq.). The mixture was heated to 150 °C for two days. The dark crude product was finely dispersed in ethyl acetate. The suspension was refluxed for 1 h. It was then cooled to room temperature and filtered. The tan product was further purified by dissolving it in $CH_2Cl_2$, followed by precipitation with ethyl acetate. Colorless needles were washed with ethyl acetate, $Et_2O$, and dried in vacuo to give $1^{sa}$ with 64% yield (1.10 g). The analytical data corresponded to those in the literature [52].

### 3.3. Synthesis and Characterization of Benzimidazolium Salt $1^{benz}$

An ACE pressure tube was loaded with 2,6-dibromopyridine (969 mg, 4.09 mmol, 1.0 eq.) and 1-(2,6-diisopropylphenyl)benzimidazole (2.30 g, 8.25 mmol, 2.0 eq.). The mixture was heated to 220 °C for three days. The product was purified by fractionate precipitation: it was dissolved in $CH_2Cl_2$, and $Et_2O$ was added dropwise, until a brown precipitate formed. The mother liquor was decanted. Further addition of more $Et_2O$ induced the precipitation of the product. Drying in vacuo afforded $1^{benz}$ as a colorless solid with 71% yield (2.30 g).

$^1$H NMR (270 MHz, DMSO-$D_6$) δ = 11.21 (s, 2 H, Benzimidazolium-**H**), 8.84 (t, *J* = 8.0 Hz, 1H, Ar**CH**), 8.74 (d, *J* = 8.0 Hz, 2H, Ar**CH**), 8.59 (d, *J* = 8.0 Hz, 2H, Ar**CH**), 7. 95–7.75 (m, 6H, Ar**CH**), 7. 68–7.59 (m, 6H, Ar**CH**), 2.41 (spt, *J* = 6.8 Hz, 4H, **CH**(CH$_3$)$_2$), 1.21 (d, *J* = 6.8 Hz, 12H, **CH**$_3$), 1.06 (d, *J* = 6.5 Hz, 12H, **CH**$_3$) ppm. $^{13}$C NMR (68 MHz, DMSO-$D_6$) δ = 146.1 (Ar**C**), 145.9 (Ar**C**), 144.6 (Ar**C**), 143.4 (Ar**CH**/Benzimidazolium **CH**), 133.1 (Ar**CH**/Benzimidazolium **CH**), 132.4 (Ar**CH**), 129.3 (Ar**C**), 128.8

(Ar**CH**), 128.7 (Ar**CH**), 127.1 (Ar**C**), 125.2 (Ar**CH**), 119.0 (Ar**CH**), 116.3 (Ar**CH**), 113.7 (Ar**CH**), 28.0 (**CH**(CH$_3$)$_2$/CH(**CH$_3$**)$_2$), 24.2 (**CH**(CH$_3$)$_2$/CH(**CH$_3$**)$_2$), 23.8 (**CH**(CH$_3$)$_2$/CH(**CH$_3$**)$_2$) ppm. UHR ESI–MS: *m/z* calculated for ([C$_{43}$H$_{46}$N$_5$$^+$]) 632.3748, found 632.3743. Melting point: 211–213 °C.

### 3.4. Synthesis and Characterization of Free Carbene 2$^{sa}$

A solution of KN(SiMe$_3$)$_2$ (117 mg, 0.585 mmol, 2.0 eq.) in toluene was added dropwise to a suspension of **1$^{sa}$** (204 mg, 0.292 mmol, 1.0 eq.) in toluene at −40 °C. The suspension was allowed to warm to room temperature, then it was filtered, and the volatiles were removed in vacuo. The colorless product was obtained in quantitative yield (154 mg). In case of the presence of adventitious water, the product was purified by washing with hexanes.

$^1$H NMR (400 MHz, C$_6$D$_6$) δ = 8.10 (d, *J* = 8.0 Hz, 2H, Ar**CH**), 7.27–7.13 (m, 7H, Ar**CH**, superimposed by solvent), 3.97–3.91 (m, 4H, **CH$_2$**), 3.51–3.44 (m, 4H, **CH$_2$**), 3.17 (spt, *J* = 6.9 Hz, 4H, **CH**(CH$_3$)$_2$), 1.28 (d, *J* = 6.9 Hz, 12H, **CH$_3$**), 1.21 (d, *J* = 7.1 Hz, 12H, **CH$_3$**) ppm. $^{13}$C NMR (101 MHz, C$_6$D$_6$) δ = 243.8 (**C** carbene), 156.2 (Ar**C**), 147.2 (Ar**C**), 139.5 (Ar**C**), 139.2 (Ar**CH**), 129.0 (Ar**CH**), 124.5 (Ar**CH**), 106.2 (Ar**CH**), 54.5 (**CH$_2$**), 46.5 (**CH$_2$**), 29.2 (**CH**(CH$_3$)$_2$/CH(**CH$_3$**)$_2$), 25.5 (**CH**(CH$_3$)$_2$/CH(**CH$_3$**)$_2$), 24.2 (**CH**(CH$_3$)$_2$/CH(**CH$_3$**)$_2$) ppm.

### 3.5. Synthesis and Characterization of Free Carbene 2$^{benz}$

A solution of KN(SiMe$_3$)$_2$ (56 mg, 0.28 mmol, 2.0 eq.) in benzene was added dropwise to a suspension of **1$^{benz}$** (112 mg, 0.14 mmol, 1.0 eq.) in benzene. The suspension was stirred for 10 min and filtered, and the volatiles were removed in vacuo to afford **2$^{benz}$** as a colorless solid in quantitative yield (89 mg). In case of the presence of adventitious water, the product was purified by washing with hexanes.

$^1$H NMR (270 MHz, C$_6$D$_6$) δ = 8.78–8.66 (m, 2 H, Ar**CH**), 8.45 (d, *J* = 7.9 Hz, 2H, Ar**CH**), 7.31–7.21 (m, 3H, Ar**CH**), 7.13 (s, 2H, Ar**CH**), 7.05 (s, 2H, Ar**CH**), 6.89–6.79 (m, 4H, Ar**CH**), 6.64 (m, 2H, Ar**CH**), 2.63 (spt, *J* = 6.9 Hz, 4H, **CH**(CH$_3$)$_2$), 1.11 (d, *J* = 6.9 Hz, 12H, **CH$_3$**), 0.84 (d, *J* = 6.9 Hz, 12 H **CH$_3$**) ppm. $^{13}$C NMR (68 MHz, C$_6$D$_6$) δ = 231.8 (**C** carbene), 154.8 (Ar**C**), 147.5 (Ar**C**), 140.7 (Ar**CH**), 139.0 (Ar**C**), 135.9 (Ar**C**), 133.6 (Ar**C**), 130.1 (Ar**CH**), 124.6 (Ar**CH**), 124.1 (Ar**CH**), 123.8 (Ar**CH**), 123.8 (Ar**CH**), 116.8 (Ar**CH**), 111.4 (Ar**CH**), 29.2 (**CH**(CH$_3$)$_2$/CH(**CH$_3$**)$_2$), 25.2 (**CH**(CH$_3$)$_2$/CH(**CH$_3$**)$_2$), 23.9 (**CH**(CH$_3$)$_2$/CH(**CH$_3$**)$_2$) ppm.

### 3.6. Synthesis and Characterization of Palladium Complex 3$^{sa}$

**Method A**: The bisimidazolinium salt **1$^{sa}$** (551 mg, 0.79 mmol, 1.0 eq.) and Pd(OAc)$_2$ (195 mg, 0.87 mmol, 1.1 eq.) were dissolved in DMSO. The dark red solution was stirred for three weeks at room temperature. The solvent was removed under reduced pressure at elevated temperatures (100 °C). The solid was dissolved in CH$_2$Cl$_2$ and filtered over diatomaceous earth. The complex was precipitated by the addition of Et$_2$O. The yellow solid was dried in vacuo overnight at 190 °C (585 mg, 92%, impure, the undefined side products in considerable amounts were identified in the $^1$H NMR spectrum).

**Method B**: The bisimidazolinium salt **1$^{sa}$** (70 mg, 0.10 mmol, 1.0 eq.) was suspended in toluene. A solution of KN(SiMe$_3$)$_2$ (40 mg, 0.20 mmol, 2.0 eq.) in toluene was added dropwise at −40 °C. The suspension was allowed to warm to room temperature and was filtered, and the volatiles were removed in vacuo. THF was added, and the solution was added to a solution of MgBr$_2$ (18 mg, 0.10 mmol, 1.0 eq.) in THF. The mixture was stirred for 10 min. [Pd(COD)Br$_2$] (37 mg, 0.10 mmol, 1.0 eq.) was added, and the suspension was stirred for additional 16 h. The solvent was removed in vacuo and washed with benzene. The product was extracted with CH$_2$Cl$_2$ and precipitated by the addition of Et$_2$O. The yellow solid was dried in vacuo to give **3$^{sa}$** in quantitative yield (80 mg).

$^1$H NMR (600 MHz, DMSO-D$_6$) δ = 8.26 (t, *J* = 8.0 Hz, 1H, Ar**CH**), 7.31–7.24 (m, 2H, Ar**CH**), 7.12 (d, *J* = 7.5 Hz, 4H, Ar**CH**), 7.05 (d, *J* = 8.1 Hz, 2H, Ar**CH**), 4.40–4.29 (m, 4H, **CH$_2$**), 4.23–4.23 (m, 4H, **CH$_2$**), 2.85 (spt, *J* = 6.4 Hz, 4H, **CH**(CH$_3$)$_2$), 1.15 (d, *J* = 6.1 Hz, 24H, **CH$_3$**) ppm. $^{13}$C NMR (151 MHz, DMSO-D$_6$) δ = 190.8 (**C** carbene), 152.3 (Ar**C**), 144.9 (Ar**CH**), 144.3 (Ar**C**), 133.8 (Ar**C**), 129.4 (Ar**CH**), 123.9 (Ar**CH**),

102.6 (Ar**C**H), 57.3 (**C**H$_2$), 43.4 (**C**H$_2$), 27.7 (**C**H(CH$_3$)$_2$/CH(CH$_3$)$_2$), 24.6 (CH(**C**H$_3$)$_2$/CH(CH$_3$)$_2$), 23.7 (CH(**C**H$_3$)$_2$/CH(CH$_3$)$_2$) ppm. UHR ESI–MS: *m/z* calculated for ([C$_{35}$H$_{45}$N$_5$PdBr$^+$]) 722.1888, found 722.1859. Melting point: >290 °C.

### 3.7. Synthesis and Characterization of Palladium Complex **3**$^{benz}$

**Method A**. The compounds **1**$^{benz}$ (544 mg, 0.69 mmol, 1.0 eq.) and Pd(OAc)$_2$ (154 mg, 0.69 mmol, 1.0 eq.) were dissolved in DMSO and stirred for 24 h at room temperature, during which a yellow precipitate formed. The suspension was heated to 40 °C for additional 24 h. The solvent was removed under reduced pressure at elevated temperatures (100 °C). The residue was dissolved in CH$_2$Cl$_2$ and filtered over diatomaceous earth. The yellow product was precipitated by the addition of Et$_2$O and dried in vacuo to give **3**$^{benz}$ with 58% yield (362 mg).

**Method B**. The compound **1**$^{benz}$ (51 mg, 0.064 mmol, 1.0 eq.) was suspended in benzene. A solution of KN(SiMe$_3$)$_2$ (25 mg, 0.13 mmol, 2.0 eq.) in benzene was added dropwise. The suspension was stirred for 10 min and filtered, and the volatiles were removed in vacuo. The residue was dissolved in THF, and the solution was added to a solution of MgBr$_2$ (12 mg, 0.065 mmol, 1.0 eq.) in THF. The mixture was stirred for 10 min. [Pd(COD)Br$_2$] (24 mg, 0.064 mmol, 1.0 eq) was added. The suspension was stirred for 16 h. Then, it was evaporated to dryness and washed with benzene. The residue was extracted with CH$_2$Cl$_2$, and the product was precipitated by Et$_2$O. The yellow solid was dried in vacuo to give **3**$^{benz}$ with 85% yield (49 mg).

$^1$H NMR (270 MHz, DMSO-D$_6$) δ = 8.79–8.63 (m, 5H, Ar**C**H), 7.78 (t, *J* = 7.6 Hz, 2H, Ar**C**H), 7.65–7.48 (m, 4H, Ar**C**H), 7.33 (d, *J* = 7.7 Hz, 4H, Ar**C**H), 7.14 (d, *J* = 8.0 Hz, 2H, Ar**C**H), 2.30 (spt, *J* = 6.8 Hz, 4H, **C**H(CH$_3$)$_2$), 1.11 (d, *J* = 6.8 Hz, 12H, **C**H$_3$), 0.88 (d, *J* = 6.8 Hz, 12H, **C**H$_3$) ppm. $^{13}$C NMR (68 MHz, DMSO-D$_6$) δ = 178.2 (**C** carbene), 151.0 (Ar**C**), 146.4 (Ar**C**H), 144.9 (Ar**C**), 135.4 (Ar**C**), 130.8 (Ar**C**H), 130.4 (Ar**C**), 129.5 (Ar**C**), 127.2 (Ar**C**H), 126.8 (Ar**C**H), 124.2 (Ar**C**H), 113.8 (Ar**C**H), 113.2 (Ar**C**H), 110.8 (Ar**C**H), 28.0 (**C**H(CH$_3$)$_2$/CH(CH$_3$)$_2$), 24.2 (CH(**C**H$_3$)$_2$/CH(CH$_3$)$_2$), 23.1 (CH(**C**H$_3$)$_2$/CH(CH$_3$)$_2$) ppm. UHR ESI–MS: *m/z* calculated for ([C$_{43}$H$_{45}$N$_5$PdBr$^+$]) 818.1890, found 818.1858. Melting point: >290 °C.

### 3.8. Synthesis and Characterization of Magnesium Complex **4**$^{sa}$

**Method A**: The bisimidazolinium salt **1**$^{sa}$ (192 mg, 0.28 mmol, 1.0 eq.) was suspended in benzene. A solution of Mg[N(SiMe$_3$)$_2$]$_2$ (95 mg, 0.28 mmol, 1.0 eq.) in benzene was added dropwise. The mixture was stirred for 16 h. The precipitate was collected, washed with pentane, and dried in vacuo to give a colorless solid with 62% yield (122 mg). Further 38% yield (79 mg) was obtained by precipitation from the mother liquor with pentane.

**Method B**: The bisimidazolinium salt **1**$^{sa}$ (93 mg, 0.13 mmol, 1.0 eq.) was suspended in toluene. A solution of KN(SiMe$_3$)$_2$ (53 mg, 0.26 mmol, 2.0 eq.) in toluene was added dropwise at −40 °C. The mixture was stirred, allowed to warm to room temperature, and filtered. The volatiles were removed in vacuo, and THF was added. The solution was added to a solution of MgBr$_2$ (25 mg, 0.13 mmol, 1.0 eq.) in THF and stirred for 1 h. The solvent was evaporated. The colorless solid was dried in vacuo and obtained in quantitative yield (95 mg).

$^1$H NMR (400 MHz, Pyridine-D$_5$) δ = 7.71 (t, *J* = 8.1 Hz, 1H, Ar**C**H), 7.37–7.26 (m, 2H, Ar**C**H), 7.17–7.06 (m, 6H, superimposed by solvent, Ar**C**H), 6.49 (d, *J* = 8.2 Hz, 2H, Ar**C**H), 4.13–4.04 (m, 4H, **C**H$_2$), 3.96–3.91 (m, 4H, **C**H$_2$), 3.68 (spt, *J* = 6.3 Hz, 4H, **C**H(CH$_3$)$_2$), 1.24 (d, *J* = 6.3 Hz, 12H, **C**H$_3$), 1.05 (d, *J* = 6.9 Hz, 12H, **C**H$_3$) ppm. $^{13}$C NMR (101 MHz, Pyridine-D$_5$) δ = 221.3 (**C** carbene), 151.5 (Ar**C**), 147.9 (Ar**C**H), 142.8 (Ar**C**H), 137.3 (Ar**C**), 130.0 (Ar**C**), 129.2 (Ar**C**H), 125.1 (Ar**C**H), 103.8 (Ar**C**H), 57.1 (**C**H$_2$), 45.6 (**C**H$_2$), 28.4 (**C**H(CH$_3$)$_2$/CH(CH$_3$)$_2$), 26.9 (CH(**C**H$_3$)$_2$/CH(CH$_3$)$_2$), 25.0 (CH(**C**H$_3$)$_2$/CH(CH$_3$)$_2$) ppm. Two signals were superimposed by the solvent signals. UHR ESI–MS: *m/z* calculated for ([C$_{35}$H$_{46}$N$_5$$^+$]) 536.3748, found 536.3731. Melting point: 220 °C, decomposition.

### 3.9. Synthesis and Characterization of Magnesium Complex *4*<sup>benz</sup>

**Method A**: The bisbenzimidazolium salt **1**<sup>benz</sup> (79 mg, 0.10 mmol, 1.0 eq.) was suspended in benzene. A solution of Mg[N(SiMe$_3$)$_2$]$_2$ (35 mg, 0.10 mmol, 1.0 eq.) in benzene was added. The mixture was stirred for 24 h, the precipitate was collected, washed with pentane, and dried in vacuo to obtain an off-white solid with 76% yield (62 mg). Further 24% yield (19 mg) was obtained by precipitation from the mother liquor with pentane.

**Method B**: The bisbenzimidazolium salt **1**<sup>benz</sup> (57 mg, 0.07 mmol, 1.0 eq.) was suspended in benzene. A solution of KN(SiMe$_3$)$_2$ (36 mg, 0.14 mmol, 2.0 eq.) in benzene was added. The mixture was stirred for 10 min and filtered. The volatiles were removed in vacuo, and THF was added. The solution was added to a solution of MgBr$_2$ (13 mg, 0.07 mmol, 1.0 eq) in THF and stirred for 1 h. The solvent was removed in vacuo, the residue was washed with pentane and dried in vacuo to give the product as an off-white solid in quantitative yield (45 mg).

Note: The NMR spectra indicated that two species formed in a ratio of 4:1. The formation of these two species was observed in various solvents (pyridine-D$_5$, THF-D$_8$, THF, or benzene). Subsequent transmetalation with [Pd(COD)Br$_2$], however, yielded only one complex. We, hence, assigned these products to a mixture of two magnesium complexes. $^1$H NMR (270 MHz, Pyridine-D$_5$) δ = 8.91 (d, *J* = 7.9 Hz, 2H, Ar**CH**, Species a/b), 8.65 (d, *J* = 8.4 Hz, 2H Ar**CH**, Species a/b), 8.11–8.27 (m, 2H, Ar**CH**, Species a/b), 8.01 (t, *J* = 7.9 Hz, 1H, Ar**CH**, Species a/b), 7.61–7.73 (m, 2H, Ar**CH**, Species a/b), 7.44–7.57 (m, 5H, Ar**CH**, Species a/b), 7.25–7.42 (m, 6H, Ar**CH**, Species a/b), 7.00–7.14 (m, 2H, Ar**CH**, Species a/b), 3.38 (spt, *J* = 6.7 Hz, 1H, **CH**(CH$_3$)$_2$, Species b), 2.75 (spt, *J* = 6.4 Hz, 4H, **CH**(CH$_3$)$_2$, Species a), 1.32 (d, *J* = 6.4 Hz, 4H, **CH**$_3$, Species b), 1.23 (d, *J* = 6.9 Hz, 12H, **CH**$_3$, Species a), 1.03 (d, *J* = 6.9 Hz, 12H, **CH**$_3$, Species a), 0.87 (d, *J* = 6.4 Hz, 4H, **CH**$_3$, Species b) ppm. $^{13}$C NMR (68 MHz, Pyridine-D$_5$) δ = 231.2 (**C** carbene), 154.7 (Ar**CH**/Ar**C**), 148.2 (Ar**CH**/Ar**C**), 147.5 (Ar**CH**/Ar**C**), 143.6 (Ar**CH**/Ar**C**), 141.3 (Ar**CH**/Ar**C**), 139.1 (Ar**CH**/Ar**C**), 139.0 (Ar**CH**/Ar**C**), 135.7 (Ar**CH**/Ar**C**), 134.2 (Ar**CH**/Ar**C**), 133.6 (Ar**CH**/Ar**C**), 131.6 (Ar**CH**/Ar**C**), 131.1 (Ar**CH**/Ar**C**), 130.5 (Ar**CH**/Ar**C**), 129.2 (Ar**CH**/Ar**C**), 125.5 (Ar**CH**/Ar**C**), 125.2 (Ar**CH**/Ar**C**), 125.0 (Ar**CH**/Ar**C**), 124.5 (Ar**CH**/Ar**C**), 124.1 (Ar**CH**/Ar**C**), 123.1 (Ar**CH**/Ar**C**), 116.6 (Ar**CH**/Ar**C**), 115.0 (Ar**CH**/Ar**C**), 114.9 (Ar**CH**/Ar**C**), 114.3 (Ar**CH**/Ar**C**), 111.9 (Ar**CH**/Ar**C**), 111.6 (Ar**CH**/Ar**C**), 29.2 (**CH**(**CH**$_3$)$_2$/**CH**(**CH**$_3$)$_2$), 28.7 (**CH**(**CH**$_3$)$_2$/**CH**(**CH**$_3$)$_2$), 26.2 (**CH**(**CH**$_3$)$_2$/**CH**(**CH**$_3$)$_2$), 25.1 (**CH**(**CH**$_3$)$_2$/**CH**(**CH**$_3$)$_2$), 24.8 (**CH**(**CH**$_3$)$_2$/**CH**(**CH**$_3$)$_2$), 24.0 (**CH**(**CH**$_3$)$_2$/**CH**(**CH**$_3$)$_2$) ppm. The second carbene signal was not observed because of the low concentration of the second compound. UHR ESI–MS: *m/z* calculated for ([C$_{43}$H$_{46}$N$_5$$^+$]) 632.3748, found 632.3746. Melting point: >250 °C.

### 3.10. Synthesis and Characterization of Iron Complex *5*<sup>sa</sup>

The free carbene (91 mg, 0.15 mmol, 1.0 eq., contained 1 eq. benzene) was dissolved in THF. MgBr$_2$ (27.3 mg, 0.15 mmol, 1.0 eq.) was added. The mixture was stirred for 10 min, and FeBr$_2$ (31.9 mg, 0.15 mmol, 1.0 eq.) was added. The suspension was stirred for 16 h. The suspension was filtered and washed with copious amounts of Et$_2$O. The remaining red-purple/pink solid was dried in vacuo to give **5**<sup>sa</sup> with 98% yield (110 mg). The purity of the product was verified by reduction with Na/Hg, as reported in the literature [52].

### 3.11. Effects of Metal Cation on Transmetalation

**General Procedure.** Each procedure was performed on a 30 mg scale. A solution of M[N(SiMe$_3$)$_2$]$_x$ (Li, K: 2.0 eq., *x* = 2; Mg 1.0 eq., *x* = 1) in benzene/THF was added dropwise to a suspension of the bisimidazolinium salt **1**<sup>sa</sup> (1.0 eq.) in benzene/THF. [Pd(COD)Cl$_2$] was added after 1 h. The reaction mixture was stirred for 16 h, and the solvent was evaporated (Scheme 6). The product selectivity was determined by $^1$H NMR spectroscopy in DMSO-D$_6$ using pyridine (8 μL) as an internal standard.

**Scheme 6.** Different reaction conditions, which were evaluated in order to determine the impact of metal cations and solvents on the crude yield of palladium complex **3$^{sa}$**.

## 4. Conclusions

We reported that transmetalation from magnesium complexes with pincer-type NHC ligands is a convenient method to synthesize related palladium and iron complexes. Of particular note, the method is also suitable for complexes with imidazolidine (saturated) NHC ligands, which cannot be obtained by other routes. Transmetalation from the lithium–carbene complex or reaction with the free carbene led, in these cases, only to mixtures which are difficult to purify and produce low yields. Furthermore, the transmetalation from the magnesium–carbene complex evolved to be superior also to other commonly applied and well-established routes, such as the transmetalation using silver(I) oxide and the in situ metalation with palladium acetate in DMSO. We, hence, conclude that transmetalation from magnesium–NHCs shows promise as a general, convenient, selective, and high-yielding synthetic approach towards transition-metal complexes with chelating NHC ligands. Further work will be directed at exploring magnesium complexes with other π-electron-deficient carbenes for transmetalation and exploiting magnesium–NHCs as transmetalating reagents for f-block and p-block elements. Preliminary experiments with Co, Ni, and p-block metals such as Pb and Bi indicate that this procedure should be indeed a versatile route to oligodentate carbene complexes.

**Supplementary Materials:** The following are available online at http://www.mdpi.com/2304-6740/7/5/65/s1, NMR spectra and crystallographic details, the CIF and the checkCIF files.

**Author Contributions:** J.M. and A.G. performed the experiments. P.S. synthesized Mg(N(SiMe$_3$)$_2$)$_2$. L.S. measured the UHR ESI–MS data. F.W.H. performed the determination of the solid-state structures. D.M. supervised the project and wrote the manuscript.

**Funding:** D.M. thanks the Fonds der Chemischen Industrie im Verband der Chemischen Industrie for a Liebig fellowship. We also thank the Dr. Hertha und Helmut Schmauser Stiftung and the German-American Fulbright Commission for financial support.

**Acknowledgments:** Continuous support by Karsten Meyer is gratefully acknowledged. We thank Ivana Ivanović-Burmazović for mass spectrometry and also thank Romano Dorta and Sjoerd Harder for their help.

**Conflicts of Interest:** The authors declare no conflict of interest.

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
