# Peer review of "Transmetalation from Magnesium–NHCs—Convenient Synthesis of Chelating π-Acidic NHC Complexes"

_inorganics, doi:10.3390/inorganics7050065_

Round 1
Reviewer 1 Report
This manuscript describes the synthesis and characterization of new magnesium complexes with pincer-type NHC ligands and their use as highly efficient transmetalation agents to synthesize the related palladium and iron complexes. The paper is well written and the experimental results are clearly presented. In my opinion, this study is interesting for organometallic synthetic chemists and deserves publication in Inorganics as it is.
Author Response
We thank the reviewer for his report and that he likes our manuscript.
Reviewer 2 Report
Dear authors, I have read your manuscript (ID number inorganics-510360) with a great pleasure. I believe it merits the publication in Inorganics. I have no found weak places and now I have to say two moments only: 1. You say that " Preliminary investigations with nickel(II) bromide, cobalt(II) bromide, bismuth(III) chloride, and lead(II) bromide indicate that the transmetallation is also feasible with these metal precursors." I hope the full results of these investigations will be presented as soon as possible! 2. As I understand, compounds 3sa and 3benz are ionic compounds of the common type [Pd(C,N,C'-carbene)Br]+Br-. In this case, the figure 2 does not show the solid-state structures of 3benz. It shows the structure of metal-containing cation [Pd(C,N,C'-carbene)Br]+ of this complex. The caption to figure 2 should be corrected. It may be made on the stage of proofs corrections. Good luck with your investigations!Author Response
Dear reviewer,
we are happy that you like our manuscript and we thank you for your suggestions. We have revised Figure 2 to include the halide anion in the revision and hope to report on our results with bismuth and lead soon.